# The ATF2/miR-3913-5p/CREB5 axis is involved in the cell proliferation and metastasis of colorectal cancer

Weiyu Dai[1,2,8], Linjie Hong[1,8], Wushuang Xiao[1], Luyu Zhang[1], Weihong Sha[2], Zhen Yu[1], Xuehua Liu[1,3], Side Liu[1,4], Yizhi Xiao[5], Ping Yang[1], Ying Peng[1], Jieming Zhang[1], Jianjiao Lin[4], Xiaosheng Wu[1], Weimei Tang[1], Zhizhao Lin[1], Li Xiang[4], Jiaying Li [1,6✉], Miaomiao Pei [1,7✉] & Jide Wang [1,4✉]

Various miRNAs have been shown to participate in the tumor progression and development of colorectal cancer (CRC). However, the role of miR-3913-5p in CRC are yet to be clearly defined. In the present study, we determine that miR-3913-5p is downregulated in CRC cell lines and CRC tissues. Exogenous miR-3913-5p expression weakens the CRC cells growth, migration and invasion. Mechanistically, miR-3913-5p directly targets the 3'UTR of CREB5. Overexpression of CREB5 reverses the suppression of CRC cells proliferation, migration and invasion induced by miR-3913-5p. Furthermore, ATF2 negatively regulates the transcription of miR-3913-5p by binding to its promoter. CREB5 can cooperate with ATF2. CREB5 is required for ATF2 in regulating miR-3913-5p. Finally, inverse correlations can be found between the expressions of miR-3913-5p and CREB5 or ATF2 in CRC tissues. Thus, a plausible mechanism of ATF2/miR-3913-5p/CREB5 axis regulating CRC progression is elucidated. Our findings suggest that miR-3913-5p functions as a tumor suppressor in CRC. ATF2/miR-3913-5p/CREB5 axis might be a potential therapeutic target against CRC progression.

[1] Guangdong Provincial Key Laboratory of Gastroenterology, Department of Gastroenterology, Nanfang Hospital, Southern Medical University, Guangzhou 510515, China. [2] Department of Gastroenterology, Guangdong Provincial People's Hospital (Guangdong Academy of Medical Sciences), Southern Medical University, Guangzhou 510080, China. [3] Department of Gastroenterology, Shunde Hospital, Southern Medical University, Foshan 528300, China. [4] Department of Gastroenterology, The Second Affiliated Hospital, School of Medicine, The Chinese University of Hong Kong, Shenzhen & Longgang District People's Hospital of Shenzhen, Shenzhen 518172, China. [5] Department of Gastroenterology, Fifth Affiliated Hospital of Sun Yat-sen University, Zhuhai 519000, China. [6] Department of Gastroenterology, The Key Laboratory of Advanced Interdisciplinary Studies Center, The First Affiliated Hospital of Guangzhou Medical University, Guangzhou 510120, China. [7] State Key Laboratory of Cancer Biology, National Clinical Research Center for Digestive Diseases, Xijing Hospital of Digestive Diseases, Fourth Military Medical University, Xi'an 710032, China. [8] These authors contributed equally: Weiyu Dai, Linjie Hong. ✉email: jiayingli1994@outlook.com; pmm202208@163.com; jidewang55@163.com

Colorectal cancer (CRC) remains the third most common cancer worldwide with more than 1.9 million new cases and the second leading cause of cancer death with an estimated 935,000 deaths in 2020[1]. Despite the increasing development of the diagnosis and treatment of CRC during the past decades, the survival of CRC patients remains unsatisfactory because of the limited strategies for detecting the cancer at an early stage and prognostic predictions[2]. The tumorigenesis and tumor progression of CRC are complex processes involving a variety of molecules such as proteins or non-coding RNAs, etc. Hence, the molecular mechanisms underlying the CRC progression still remain to be fully elucidated. Identifying novel important molecules in CRC progression might contribute significantly to the development of new diagnostic and therapeutic strategies.

MicroRNAs (miRNAs) are a type of small non-coding RNAs involved in post-transcriptional regulation via interacting with the 3′-untranslated region (3′-UTR) of the corresponding target mRNA[3]. miRNAs are known to play critical roles in the regulation of many biological processes, such as cell growth, migration, invasion, apoptosis and cellular differentiation[4]. A growing body of evidence has revealed that miRNAs participate in a variety of diseases, including human cancer[5,6]. miRNAs can function as regulators in CRC, and be important for CRC diagnostic and therapeutic applications[7–9]. To date, accumulating studies have indicated the certain miRNAs involved in the tumor progression of CRC. For instance, miR-934 facilitates CRC cell growth, migration, invasion and angiogenesis by targeting BTG2[10]. miR-576-5p participates in the suppression of CRC progression induced by hsa_circ_0001666[11]. miR-154-5p plays an important role in the process of the long noncoding RNA SNHG1 mediating CRC cell growth[12]. miR-3913-5p is a less well-studied miRNA. Although recent studies have indicated the critical role of miR-3913-5p in cholangiocarcinoma and non-small cell lung cancer[13,14], the biological role and the potential mechanism of miR-3913-5p in CRC progression have not been reported yet.

In this work, we demonstrated the miR-3913-5p was frequently downregulated in CRC cell lines and CRC tissues. miR-3913-5p suppressed the CRC cell growth, migration and invasion in vitro and in vivo. miR-3913-5p directly targeted the 3′UTR of CREB5. Furthermore, ATF2 negatively regulated miR-3913-5p expression by binding to its promoter. Overall, these results revealed the critical role and potential mechanism of miR-3913-5p in inhibiting CRC cell proliferation, migration and invasion, providing a novel therapeutic target for CRC treatment.

## Results

### miR-3913-5p is frequently downregulated in gastrointestinal cancers.
To the explore effect of miR-3913-5p on the outcomes of patients with cancers, we applied a bioinformatics analysis using OncomiR database (http://www.oncomir.org/, Supplementary Fig. 1). We revealed that low expression of miR-3913-5p was found to be associated with poor survival in bladder urothelial carcinoma, head and neck squamous cell carcinoma, thymoma and rectum adenocarcinoma. However, the role of miR-3913-5p in CRC has not yet been reported in PubMed database. To identify the expression of miR-3913-5p in CRC, we used qPCR to measure the miR-3913-5p expression in CRC cell lines, 91 pairs of CRC tissues and matched normal tissues. Compared with the normal colonic mucosal cell line FHC, miR-3913-5p expression was substantially reduced in all six CRC cell lines, including LoVo, SW480, Caco2, HT-29, RKO and SW1116 (Fig. 1a). Low expression of miR-3913-5p was detected in 70.3% (64 of 91) of CRC tissues compared with the corresponding nontumorous tissues (Fig. 1b, c).

To investigate the relationship between miR-3913-5p level and CRC progression, we analyzed the miR-3913-5p expression in different clinicopathological subgroups stratified according to gender, age, tumor size, differentiation, tumor invasion, lymph node metastasis, distant metastasis and TNM stage. The results showed that low expression of miR-3913-5p was associated with poor differentiation ($p = 0.035$), stage III–IV ($p < 0.001$), T3–T4 ($p = 0.002$), lymph node metastasis ($p < 0.001$), distant metastasis ($p = 0.001$) (Fig. 1d, Supplementary Table 1). Moreover, we performed ISH of colorectal mucosa samples. The results also showed that the low miR-3913-5p staining in CRC tissues (Fig. 1e). Thus, these data indicated that miR-3913-5p was frequently downregulated in CRC.

Furthermore, we examined the miR-3913-5p expression in 80 pairs of human GC tissues and matched normal gastric mucosa tissues by qPCR. miR-3913-5p expression in GC tissues was significantly lower than the corresponding normal tissues (Fig. 1f, g). What's more, statistical analysis represented a strong correlation between miR-3913-5p expression and differentiation ($p = 0.021$), tumor invasion ($p < 0.001$), lymph node metastasis ($p = 0.001$) and TNM stage ($p = 0.004$) (Fig. 1h, Supplementary Table 2).

These findings suggested the important role of miR-3913-5p in gastric and CRCs.

### miR-3913-5p inhibits CRC cell proliferation.
To elucidate whether miR-3913-5p played an important role in tumorigenesis and progression, we selected CRC cells and tissues. We next transfected miR-3913-5p mimics into LoVo and SW480 cell lines with endogenous expression of miR-3913-5p, whereas miR-3913-5p inhibitor was transfected into normal colonic mucosal cell line FHC. The transfection efficiencies were confirmed by qPCR (Fig. 2a). We assessed the CRC cell proliferation in vitro by CCK-8 assay, colony formation and EdU assays. The results of CCK-8 assays showed a significant reduction in the growth rate of CRC cells after transfected with miR-3913-5p mimics (Fig. 2b). On the contrary, there was an increase in the growth rate of miR-3913-5p inhibitor-treated FHC cells (Fig. 2c). Overexpression of miR-3913-5p could weakened the colony formation abilities of CRC cells (Fig. 2d), while treatment with miR-3913-5p inhibitor in FHC showed the opposite result (Fig. 2e). The CRC cells transfected with miR-3913-5p mimics exhibited lower proliferation rate than CRC cells transfected with m-NC (Fig. 2f). Nevertheless, the inhibition of miR-3913-5p expression could increase the EdU incorporation in FHC (Fig. 2g). What's more, Hoechst 33258 staining showed that miR-3913-5p overexpression induced apoptosis in CRC cells (Supplementary Fig. 2a).

To determine whether miR-3913-5p affected CRC cell growth in vivo, a lentivirus-based system was used and we applied a subcutaneous tumor model. LoVo cells were chosen for the lentiviral-mediated stable miR-3913-5p overexpression. The LoVo cells stably expressing Lv-miR-3913-5p or Lv-NC were injected subcutaneously into nude mice. As shown in Fig. 2h, i, the subcutaneous tumors of mice in the Lv-miR-3913-5p groups exhibited lower growth than those in the Lv-NC groups. And Ki-67 and cleaved caspase-3 staining by IHC showed less proliferation and induced apoptosis in the Lv-miR-3913-5p groups (Supplementary Fig. 2b). The results above suggested the inhibition of proliferation in CRC in vitro and in vivo by miR-3913-5p.

### miR-3913-5p suppresses CRC cell migration and invasion.
Transwell assays and wound scratch assay were performed to assess the ability of migration and invasiveness of CRC cells with miR-3913-5p overexpression or interference in vitro. In the transwell migration and invasion assays, miR-3913-5p mimics-

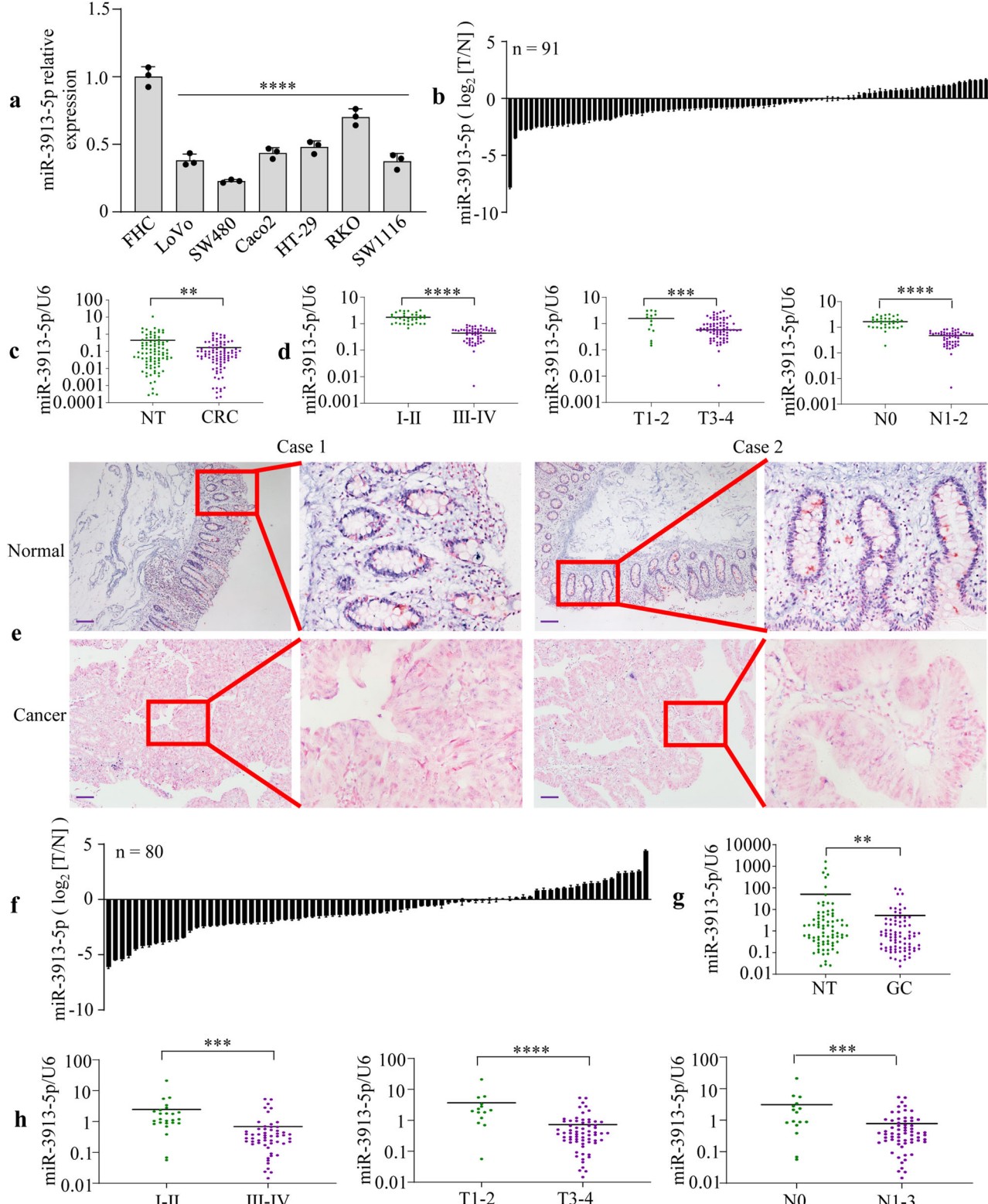

**Fig. 1 miR-3913-5p is frequently downregulated in gastrointestinal cancers. a** Decreased expression of miR-3913-5p was found in six CRC cell lines (LoVo, SW480, Caco2, HT-29, RKO and SW1116) compared with the immortalized colonic epithelial cell line FHC. One-way ANOVA and Dunnett's T3 multiple comparison test; ****$p < 0.001$. **b** QPCR analysis of miR-3913-5p expression in 91 pairs of CRC samples and corresponding normal tissues. **c** miR-3913-5p expression was lower in CRC tissues than that in the matched normal tissues. Student's $t$ test; **$p < 0.05$. **d** The miR-3913-5p expression in different subgroups based on CRC clinicopathological parameters including TNM stage, T stage and N stage. Student's $t$ test; ***$p < 0.01$; ****$p < 0.001$. **e** ISH analysis of miR-3913-5p expression pattern in CRC tissues and corresponding normal tissues. **f** The expression of miR-3913-5p in 80 paired GC tissues and the adjacent normal tissues detected by qPCR. **g** The miR-3913-5p expression level between 80 pairs of GC tissues and corresponding normal tissues. Student's $t$ test; **$p < 0.05$. **h** The miR-3913-5p expression level in the different subgroups of GC patients stratified based on TNM stage, T stage and N stage. Student's $t$ test; ***$p < 0.01$; ****$p < 0.001$. Scale bars, 100 μm in (**e**).

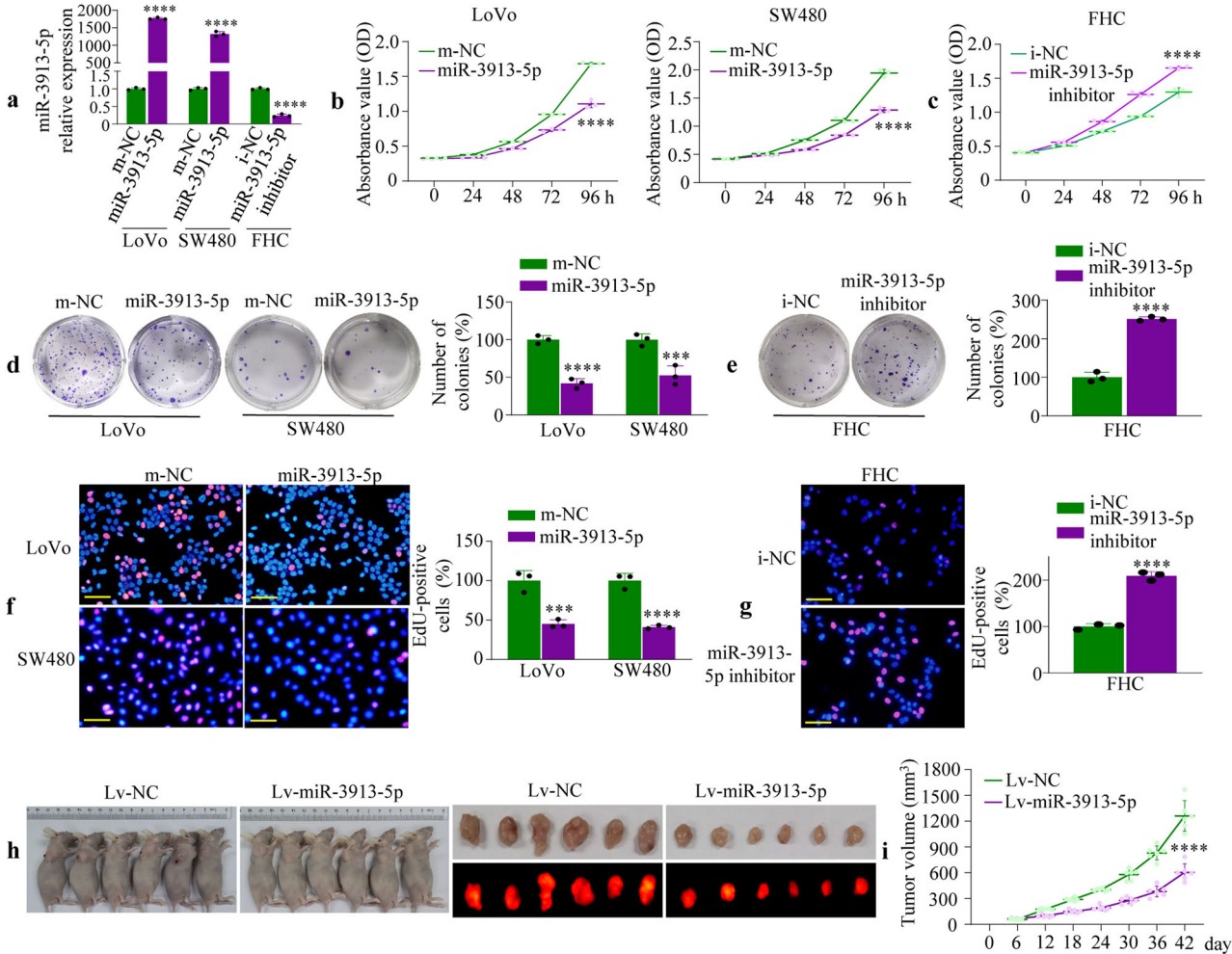

**Fig. 2 miR-3913-5p attenuates the CRC cell growth. a** The transection efficiencies of miR-3913-5p mimics or inhibitor in CRC cells or FHC cells. Student's *t* test; ****$p < 0.001$. The CCK-8 assays were performed to measure the effects of the miR-3913-5p mimics or inhibitor on the growth rate of CRC cells (**b**) or FHC cells (**c**). Student's *t* test; ****$p < 0.001$. Colony formation assays were used to determine the effects of the miR-3913-5p mimics or inhibitor on the proliferative abilities of CRC cells (**d**) or FHC cells (**e**). Quantification of crystal violet stained cell colonies formed by indicated GC cell lines, 14 days after inoculation. Student's *t* test; ***$p < 0.01$; ****$p < 0.001$, m-NC vs. miR-3913-5p. EdU assays were performed to detect the effects of the miR-3913-5p mimics or inhibitor on the DNA synthesis in CRC cells (**f**) or FHC cells (**g**). Student's *t* test; ***$p < 0.01$; ****$p < 0.001$, m-NC vs. miR-3913-5p. **h** The growth of the mice subcutaneous tumors after CRC cells subcutaneous injection. Student's *t* test; ****$p < 0.001$. **i** The representative images of the two groups (①Lv-NC; ②Lv-miR-3913-5p) of mice subcutaneous tumors resected at 42 days after CRC cells inoculation. Scale bars, 50 μm in (**f**) and (**g**).

treated CRC cells showed reduced cells migration (Fig. 3a) and invasion (Fig. 3b) as observed by crystal violet staining, whereas miR-3913-5p depletion led to an increase in the migration and invasive capacity of FHC cells (Fig. 3c, d). Wound scratch assay exhibited that miR-3913-5p overexpression inhibited the migratory activity of CRC cells (Fig. 3e). On the contrary, miR-3913-5p interference increased the cell migration of FHC cells (Fig. 3f).

To observe the effect of miR-3913-5p on CRC metastasis in vivo, we injected the LoVo cells stably expressing Lv-miR-3913-5p or Lv-NC into nude mice through the tail vein to examine their lung metastasis. The mice were euthanized and the number of metastatic lung nodules in individual mice was observed at 50 days after CRC cells injection. As shown in Fig. 3g, h, the Lv-miR-3913-5p group yielded a significantly reduced lung metastatic lesions compared with the control group. And MMP2 staining by IHC revealed less metastasis in the Lv-miR-3913-5p groups (Supplementary Fig. 3). Collectively, miR-3913-5p could suppress the CRC cell migration and invasion in vitro and in vivo.

**CREB5 is a functional target of miR-3913-5p.** To further investigate the potential mechanism of miR-3913-5p in CRC progression, we have created a Venn diagram analysis of predicted miRNA-3913-5p targets by using five independent databases: DIANA-mT (http://diana.imis.athena-innovation.gr/DianaTools/index.php?r=microT_CDS/index), mirDIP (http://ophid.utoronto.ca/mirDIP/index.jsp), miRmap (https://mirmap.ezlab.org/app/), miRPathDB (https://mpd.bioinf.uni-sb.de/overview.html) and TargetScan (http://www.targetscan.org/vert_72/). The results displayed that 39 genes were predicted target genes (Fig. 4a). Previously, CDH12[15], NRF1[16], SMAD2[17], TRPS1[18] and CREB5[19] were found to be highly expressed in CRC. Subsequently, the expression of the five target genes were measured by qRT-PCR in miR-3913-5p mimics treated or m-NC treated LoVo and SW480 cells. The results showed that CREB5 mRNA level displayed the most significant change in the CRC cells that were subjected to miR-3913-5p (Fig. 4b). Next, western blot analysis revealed that exogenous overexpression of miR-3913-5p resulted in the reduction of the CREB5 protein in LoVo and SW480 cells (Fig. 4c) and

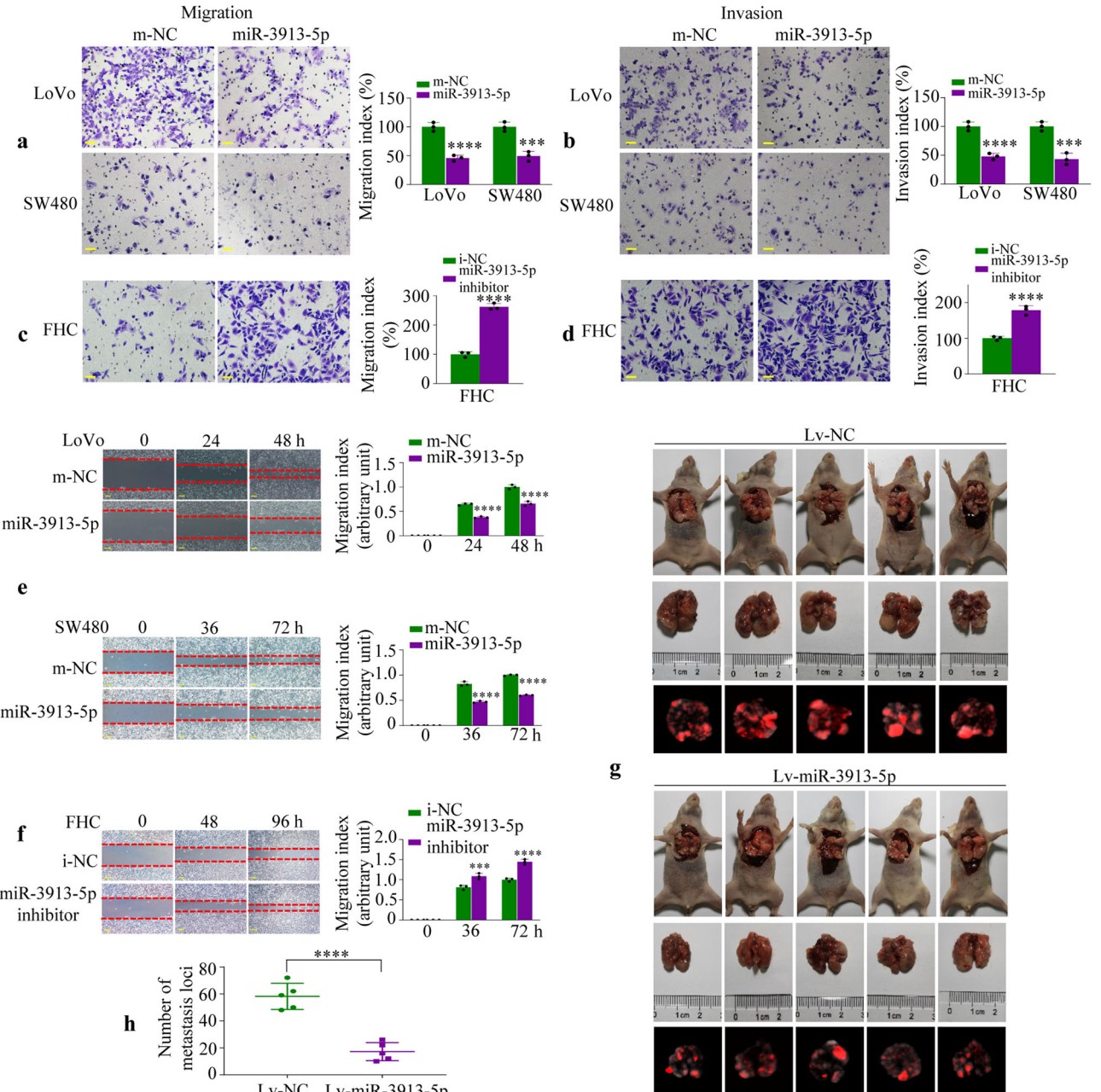

**Fig. 3 miR-3913-5p inhibits the CRC cell migration and invasion.** Transwell assays were used to detect the migration (**a**) and invasive (**b**) abilities of CRC cells after transfection with miR-3913-5p and the control. Student's *t* test; ***$p < 0.01$; ****$p < 0.001$. The migratory (**c**) and invasive (**d**) abilities of FHC cells treated with miR-3913-5p inhibitor and its control were measured by transwell assays. Student's *t* test; ****$p < 0.001$. The cell motility of miR-3913-5p mimics-treated CRC cells (**e**) or miR-3913-5p inhibitor-treated FHC cells (**f**) were detected by wound healing assays. Student's *t* test; ***$p < 0.01$; ****$p < 0.001$. **g** Representative images of the metastatic lesions in the mice lungs in the two group (①Lv-NC; ②Lv-miR-3913-5p) are shown. **h** The numbers of the lung metastatic nodules were counted. Student's *t* test; ****$p < 0.001$. Scale bars, 50 μm in (**a**–**d**), 100 μm in (**e**, **f**).

conversely, miR-3913-5p inhibition led to the increase of the CREB5 protein in FHC cells (Fig. 4d).

Then, we explored whether CREB5 was a direct target of miR-3913-5p. By using TargetScan database, two putative binding sites in the 3′UTR of CREB5 (Fig. 4e) were found. Luciferase reporter assay was used to examine whether miR-3913-5p could bind to the CREB5 3′UTR. Subsequently, we constructed wide-type CREB5 3′UTR fragments (WT1 or WT2) containing the two putative binding sites and their mutant ones (MT1 or MT2), and they were cloned into luciferase reporter vectors (Fig. 4e). These vectors and miR-3913-5p mimics were co-transfected into LoVo and SW480 cells. Significant decrease of luciferase activities was

observed in WT1 and WT2 vectors after miR-3913-5p over-expression; of note, the luciferase signal intensities were restored by mutations in the putative binding sites (Fig. 4f). These finding confirmed the direct interaction between miR-3913-5p and the CREB5 3′UTR.

To evaluate the biological role of CREB5 in miR-3913-5p-mediated proliferation, migration and invasion, we rescued CREB5 expression in miR-3913-5p-transfected CRC cells. Colony formation and EdU assays were used to assess the proliferation of the CRC cells, while transwell assays were applied to evaluate the migration and invasive capacities of the CRC cells. While miR-3913-5p overexpression resulted in a decrease of numbers and

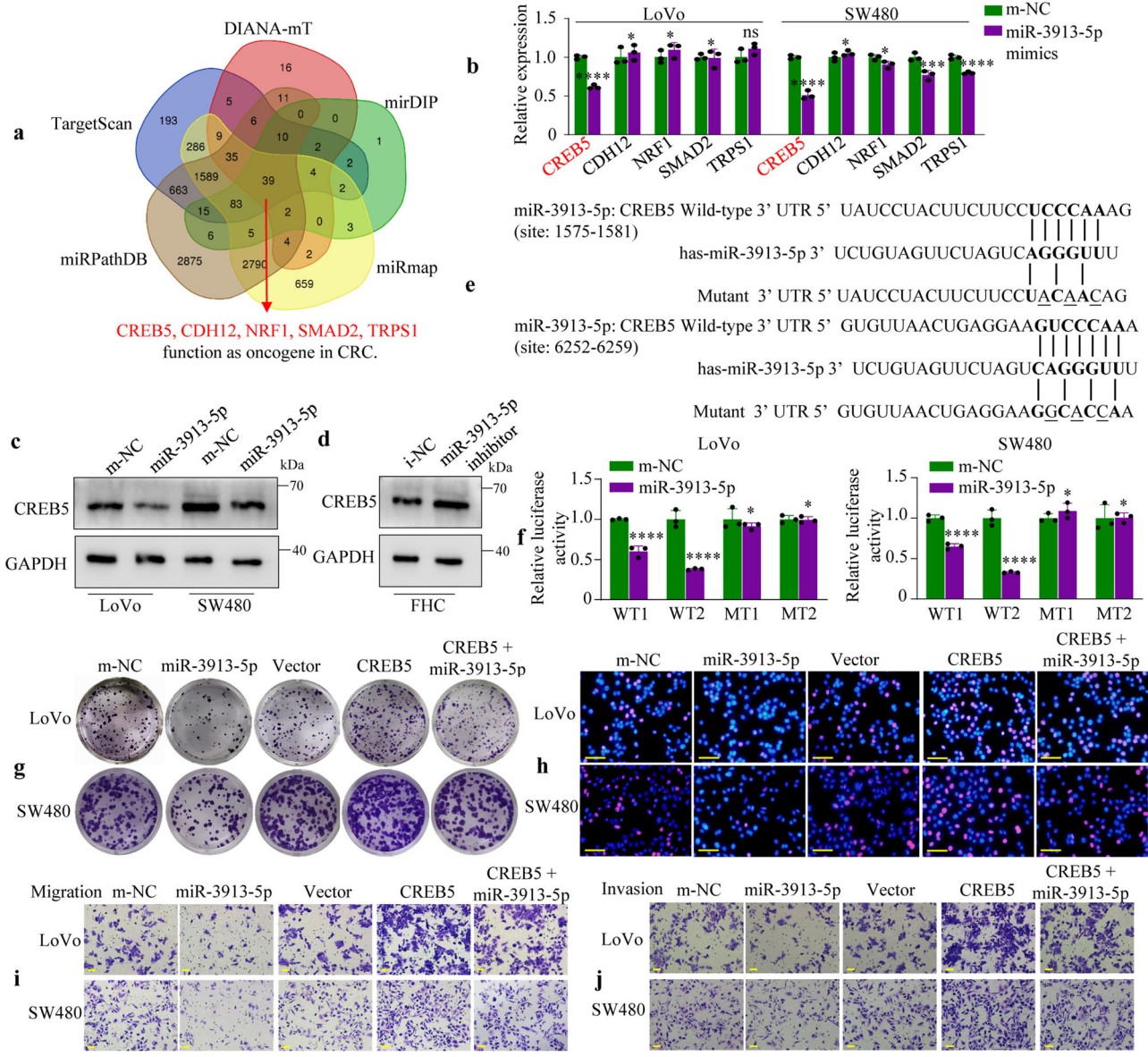

**Fig. 4 CREB5 is the target of miR-3913-5p. a** Venn diagram analyses of five independent databases revealed 39 possible targets of miR-3913-5p. **b** The screening for targets of miR-3913-5p in LoVo and SW480 cells by qPCR. Student's t test; *p > 0.05; **p < 0.05; ***p < 0.01; ****p < 0.001. Western blot assays were used to detect the CREB5 protein levels in CRC cells (**c**) or FHC cells (**d**) after transfection with miR-3913-5p mimics and m-NC, or inhibitor and i-NC. **e** The potential binding sites of miR-3913-5p in 3'UTR of CREB5 predicted by TargetScan database and the structure of the wide-type (WT1, WT2) and mutant (MT1, MT2) luciferase reporter vectors. **f** Luciferase reporter assays were performed to determine miR-3913-5p direct targeting the 3'UTR of CREB5. Student's t test; *p > 0.05; ****p < 0.001. Colony formation assays (**g**) and EdU assays (**h**) revealed that the suppression of CRC cells proliferation induced by miR-3913-5p was counteracted after administration of CREB5. Transwell assays showed that inhibition of CRC cells migratory (**i**) and invasive (**j**) abilities by miR-3913-5p was abrogated after CREB5 overexpression. Scale bars, 50 μm in (**h–j**).

sizes of CRC cell colonies and EdU incorporation, restoring CREB5 could reverse the miR-3913-5p-mediated suppression of CRC cell growth (Fig. 4g, h, Supplementary Fig. 4a, b). In addition, decrease of migratory and invasive potential in LoVo and SW480 cells could be observed following miR-3913-5p overexpression, but treatment with CREB5 restoration removed these effects on CRC cell migration and invasion (Fig. 4i, j, Supplementary Fig. 4c, d).

Taken together, these results suggested that miR-3913-5p might inhibit CRC growth and metastasis by targeting CREB5.

**ATF2 negatively regulates miR-3913-5p by binding to its promoter.** CREB5 (cAMP responsive element binding protein 5)

belongs to ATF/CREB family. Previous studies have revealed that CREB5 was a transcriptional activator and also acted as co-factors for the transcription factor (TF) FOXA1[19,20]. We hypothesized that CREB5 interacted with the ATF/CREB family of TF proteins[21]. We next determined the target-interacting proteins of CREB5 and showed that ATF2, ATF7 and BATF3 might be associated with CREB5 in the BioGRID databases (Fig. 5a). Subsequently, we tried to determine whether ATF2, ATF7 and BATF3 could regulate miR-3913-5p transcription. We then used small-interfering RNAs (siRNAp, pooled siRNA mixtures: siRNA1, siRNA2 and siRNA3) to inhibit three molecules expression in LoVo and SW480 cells. The qPCR results showed a markedly increase of miR-3913-5p expression in CRC cells after knockdown of ATF2. In contrast, the miR-3913-5p expression

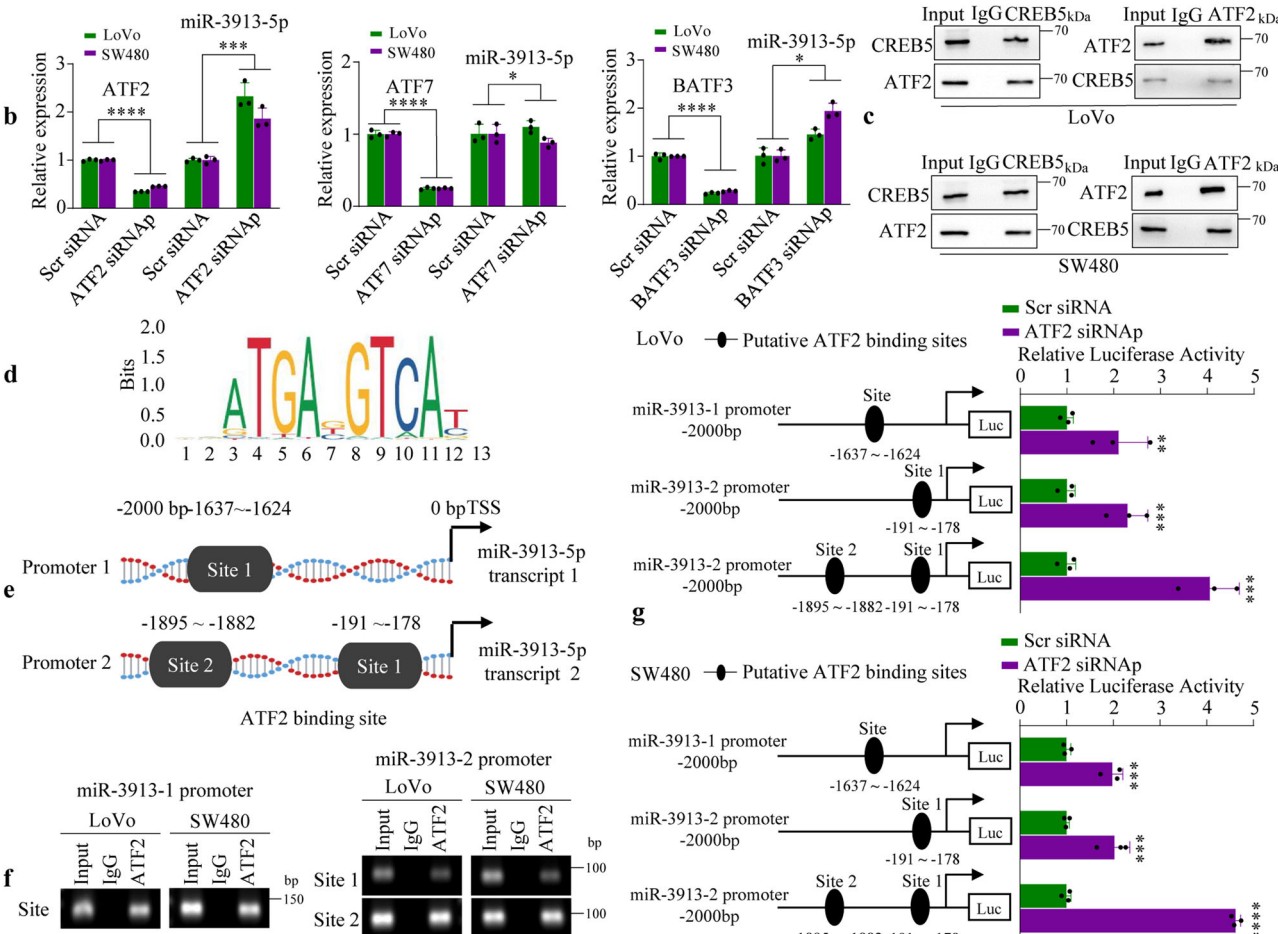

**Fig. 5 ATF2 transcriptionally regulates miR-3913-5p by binding to its promoter. a** The CREB5 interacted with transcription factor proteins. **b** QPCR analyses of miR-3913-5p expression in CRC cells following treatment of ATF2, ATF7 or BATF3 interference. Student's *t* test; *p > 0.05; ***p < 0.01; ****p < 0.001. **c** Cell lysates were immunoprecipitated by anti-CREB5 antibody or anti-ATF2 antibody. IgG was used as a control. Western blot analyses were performed using anti- CREB5 antibody or anti-ATF2 antibody. **d** The transcriptional factor ATF2-binding motif was predicted by informatics analysis. **e** Schematic illustration of the miR-3913-5p promoter with a potential ATF2 binding site. **f** Amplification of fragments containing ATF2-binding sites after ChIP assays using anti-ATF2 antibody is indicated in PCR gel. **g** Luciferase reporter assays were used to confirm the ATF2 binding to the promoter of miR-3913-5p. Student's *t* test; **p < 0.05; ***p < 0.01; ****p < 0.001.

level remained unchanged upon knockdown of ATF7 and BATF3 in LoVo and SW480 cells (Fig. 5b).

We sought to further confirm the interaction between CREB5 and ATF2 by co-immunoprecipitation assay. Co-immunoprecipitation of whole cell lysates were performed by using specific antibodies. We found that ATF2 endogenous protein were co-immunoprecipitated using an anti-CREB5 antibody, while anti-ATF2 antibody could co-immunoprecipitate CREB5 from LoVo and SW480 cell extracts, indicating that CREB5 and ATF2 interact with each other in CRC cells (Fig. 5c).

Bioinformatics showed that the miR-3913 have two transcripts including MIR3913-1 (NR_037475.1) and MIR3913-2 (NR_037476.1) (http://asia.ensembl.org/Homo_sapiens/Info/Index). To determine whether miR-3913-5p transcript could be a direct transcriptional target of ATF2, we scanned ~2-kb of the promoter region of MIR3913-1 and MIR3913-2 with the ATF2

DNA-binding consensus sequence using UCSC database (http://genome.ucsc.edu/). Subsequently, the following one potential ATF2 binding site [TF score outoff: 85%]: −1637bp to −1624bp region in the MIR3913-1 promoter and two putative binding sites: −1895bp to −1882bp and −191bp to −178bp regions in the MIR3913-2 promoter were predicted by using JASPAR database (https://jaspar.genereg.net/) (Fig. 5d, e).

A ChIP assay further confirmed that ATF2 protein was recruited to the three putative binding sites in the promoter of miR-3913-5p in LoVo and SW480 cells (Fig. 5f). Afterward, we cloned the MIR3913-1 promoter region containing putative binding site or the MIR3913-2 promoter region containing putative binding site1 or site1 + site2 into pGL3-basic vectors, and performed the dual-luciferase reporter assay by co-transfecting the vectors with ATF2 siRNAp in CRC cells. The results demonstrated that interference of ATF2 enhanced the

luciferase activities of the three binding sites in the promoter of miR-3913-5p in LoVo and SW480 cells, suggesting that all three sites were functional sites in ATF2 regulating miR-3913-5p (Fig. 5g). Together, these results revealed that transcriptional factor ATF2 bound to the promoter of miR-3913-5p to suppress its expression.

**ATF2 mediates CRC cell proliferation and metastasis through miR-3913-5p.** For the purpose of exploring whether miR-3913-5p played an important role in the function of ATF2 in regulating CRC cell proliferation and metastasis, CRC cells were co-treated with ATF2 interference and miR-3913-5p inhibition. Colony formation, EdU assay, or transwell assays were performed to examine CRC cell growth, migration, and invasion. As shown, transfection of ATF2 siRNAp decreased the cell growth activities of LoVo and SW480 cells, which was attenuated by co-treatment with the miR-3913-5p inhibitor (Fig. 6a, b). Meanwhile, it was also found that ATF2 interference inhibited the migratory and invasive cavities of LoVo and SW480 cells, which could be reversed by co-treatment with miR-3913-5p inhibition (Fig. 6c, d). Thus, these data suggested that ATF2 regulated CRC cell growth and metastasis by suppressing miR-3913-5p.

**Correlation among ATF2, miR-3913-5p and CREB5 in CRC.** Here, we could find that ATF2 regulated miR-3913-5p transcriptionally and miR-3913-5p mediated CREB5 by targeting its 3′UTR, so ATF2 and miR-3913-5p were both upstream molecules of CREB5. Then, to determine the correlation among the expression of ATF2, miR-3913-5p and CREB5 in CRC, we examined the protein expression of CREB5 and ATF2 in 12 pairs CRC and corresponding normal tissues by western blot and miR-3913-5p expression by qPCR. The results exhibited the upregulated expression of CREB5 (9/12) or ATF2 (9/12) proteins and reduced expression of miR-3913-5p (10/12) in 12 paired CRC tissues compared with the matched normal tissues (Fig. 7a, b). Subsequently, the upregulated mRNA levels of CREB5 and ATF2 in 91 pairs of CRC tissue and corresponding nontumorous tissues were further verified by qPCR (Fig. 7c, d). In addition, high expression of CREB5 or ATF2 could be frequently observed among the CRC patients with lymph node metastasis (CREB5: $p < 0.001$; ATF2: $p = 0.011$), distant metastasis (CREB5: $p < 0.001$; ATF2: $p = 0.016$) or III–IV TNM stage (CREB5: $p < 0.001$; ATF2: $p = 0.008$) (Supplementary Tables 3, 4). Based on the qPCR results of miR-3913-5p, CREB5 and ATF2 in 91 pairs of CRC tissues and matched normal tissues, pearson correlation analyses showed a positive correlation between CREB5 and ATF2, and a negative correlation between miR-3913-5p and CREB5 or ATF2 (Fig. 7e–g). The ISH staining of miR-3913-5p and IHC staining of CERB5 and ATF2 identified the lower miR-3913-5p expression and higher CERB5 or ATF2 expression in CRC tissues than in matched normal tissues in 15 cases (Fig. 7h, Supplementary Fig. 5a). Positive correlation between CREB5 and ATF2 could also be found according to their IHC staining scores (Supplementary Fig. 5b). What's more, increased expression of CERB5 and ATF2 were observed in the mice subcutaneous tumors with low miR-3913-5p expression (Supplementary Fig. 5c). Taken together, miR-3913-5p inversely correlated with CREB5 and ATF2 expression in CRC. Subsequently, the colony formation, EdU and transwell assays were conducted to explore the role of CREB5 in ATF2 regulating miR-3913-5p and investigate the function of the CREB5 and ATF2 interaction. It was showed that in LoVo and SW480 cells, miR-3913-5p reduced the strengthened proliferation, migration and invasion by ATF2, and CREB5 deteriorated these changes (Supplementary Fig. 6a–d). What's more, miR-3913-5p inhibition impaired the reduced CRC cells

growth, migratory and invasive cavities by ATF2 inhibition, while CREB5 suppression could reversed the effects above (Supplementary Fig. 7a–d). So CREB5 was required for ATF2 in regulating miR-3913-5p.

## Discussion

Though miRNAs' roles in CRC have been extensively revealed, the significance and underlying mechanism of miR-3913-5p in CRC remains unknown. Our present results highlighted the importance of upregulation of miR-3913-5p expression for CRC cells to abrogate the aggressive phenotype. By confirming the binding of miR-3913-5p to the CREB5 3′UTR and ATF2 interacting with the promoter of miR-3913-5p, the ATF2/miR-3913-5p/CREB5 axis was identified to play a vital role in CRC development, suggesting that it might serve as potential targets for managing CRC.

The roles of miR-3913-5p have been studied in other cancers. It was indicated that miR-3913-5p might be an oncogene and affected the cell growth and migration in cholangiocarcinoma[13]. What's more, exosome-derived miR-3913-5p in the peripheral blood of non-small cell lung cancer was reported to be associated with osimertinib resistance[14]. Though the relationships between miR-3913-5p and tumor progression have been characterized for the cancers above, little is known of its function and potential mechanisms in CRC. Our results discovered the reduced expression of miR-3913-5p in CRC cell lines and CRC tissues. The cancer specimens of CRC patients with poor differentiation, T3–T4, lymph node metastasis, distant metastasis or stage III–IV were more likely to exhibit low expression of miR-3913-5p. Exogenous introduction of miR-3913-5p could markedly weaken the CRC cells proliferation, migratory and invasive abilities in vitro and in vivo. In contrast to the above two studies about miR-3913-5p in cholangiocarcinoma and non-small cell lung cancer, our results indicated miR-3913-5p acted as a potential tumor suppressor in CRC. What's more, the expression profile of miR-3913-5p in GC was investigated in our study, and miR-3913-5p was first confirmed to be frequently downregulated in GC tissues, which suggested the critical role of miR-3913-5p in gastrointestinal cancers. It is worthy of further studies in lab to explore the functions and molecular mechanisms of miR-3913-5p in GC.

CREB5, the product of which belongs to the cAMP response element-binding protein family, is a transcriptional factor in eukaryotic cells[22]. Prior works have reported the roles of CREB5 in several types of cancers. CREB5 has been verified to be upregulated in hepatocellular carcinoma[23], ovarian cancer[22], and its high expression was correlated with poor prognosis. The elevated CREB5 promoted cellular proliferation in hepatocellular carcinoma, cell invasion in ovarian cancer, and resistance to androgen-receptor antagonists in prostate cancer[22–24]. CREB5 also played an important role in CRC[19], CREB5 exerted the oncogenic roles through expediting invasiveness and metastasis by upregulating MET expression to activate HGF-MET signaling. What's more, miRNAs such as miR-204, let-7a-5p, miR-132-3p and miR-125a could participate in tumor progression by targeting CREB5[25–28]. To our knowledge, the relationship between miR-3913-5p and CREB5 has not been previously documented. Our work first demonstrated that miR-3913-5p directly targeted the 3'UTR of CREB5 by luciferase reporter assays. Consistent with the previous studies, we observed that CREB5 overexpression could promote CRC cells growth, migration and invasion in vitro. Furthermore, CREB5 overexpression reversed the suppression of cellular proliferation, migratory and invasive capacities by miR-3913-5p in CRC. Moreover, miR-3913-5p expression negatively correlated with CREB5 expression in 91 CRC tissues. Our study

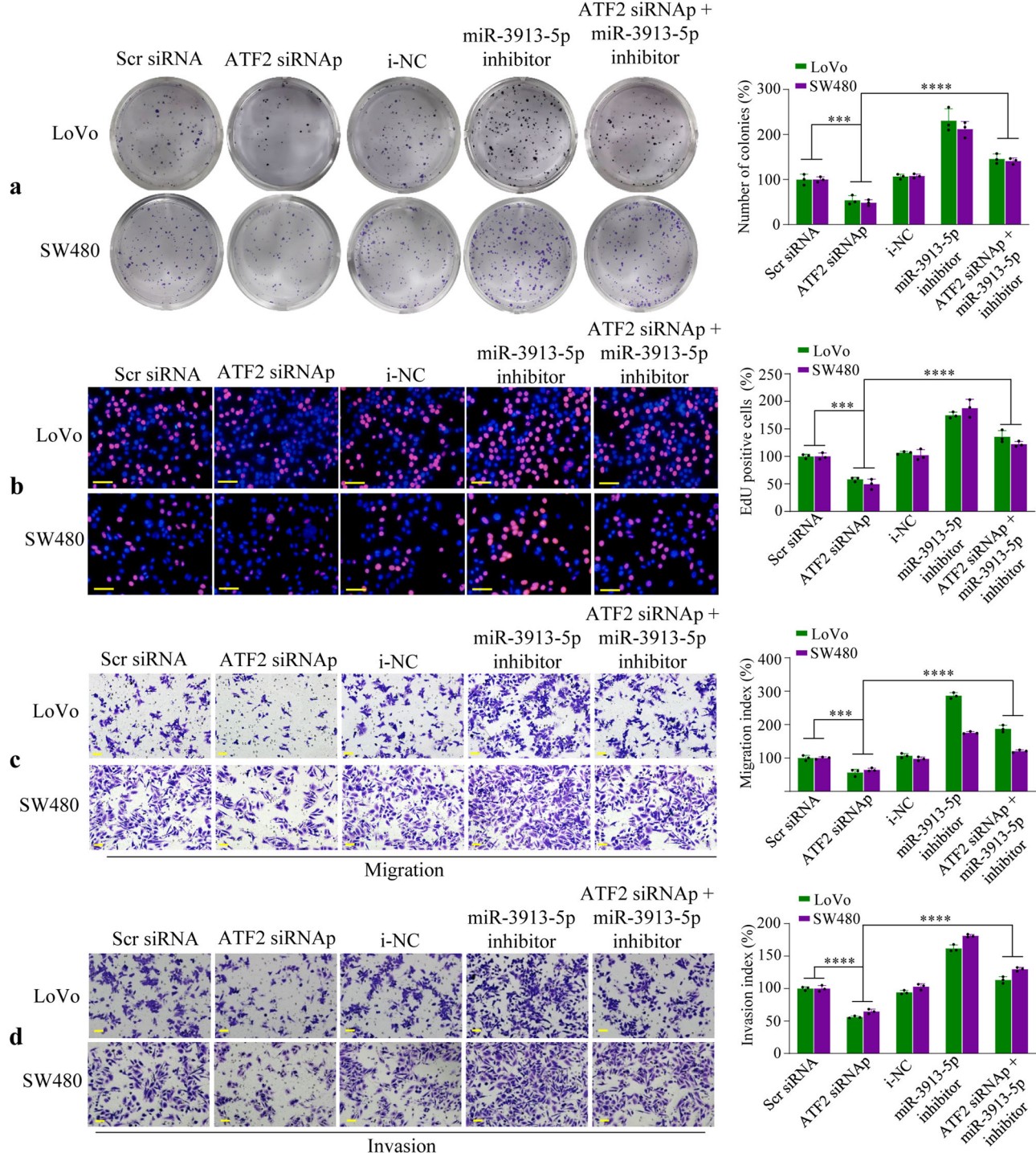

**Fig. 6 Inhibiting miR-3913-5p reverses the suppressions of CRC cell proliferation, migration and invasion induced by ATF2 interference.** Colony formation assays (**a**) and EdU assays (**b**) revealed that inhibition of miR-3913-5p weakened the suppression of CRC cells growth by ATF2 interference. Student's $t$ test; ***$p < 0.01$; ****$p < 0.001$. Transwell assays showed that the suppression of CRC cells migration (**c**) and invasion (**d**) abilities induced by ATF2 interference was reversed by miR-3913-5p inhibitor. Student's $t$ test; ***$p < 0.01$; ****$p < 0.001$. Scale bars, 50 μm in (**b–d**).

revealed the relationship between miR-3913-5p and CREB5, providing the potential mechanism of miR-3913-5p in CRC progression. Hwang et al. demonstrated that the interactions of CREB5 and FOXA1 could promote EMT signaling in AR-positive-resistant cells[20], suggesting the interactions of CREB5 with other transcriptional factors might play an important role in cancer progression. Subsequently, consistent with the previous study uncovering the interaction between CREB5 and ATF2[29], we showed that CREB5 cooperated with ATF2 in CRC cells by co-IP assays. Then, by using bioinformatics database and qPCR analyses, CREB5 expression was found to be positively correlated with ATF2 expression. These results not only indicated the interaction between CREB5 and ATF2 in CRC, but also suggested the important roles of ATF2 in CRC. Thus, we further investigated the potential mechanism of ATF2 in CRC in the following study.

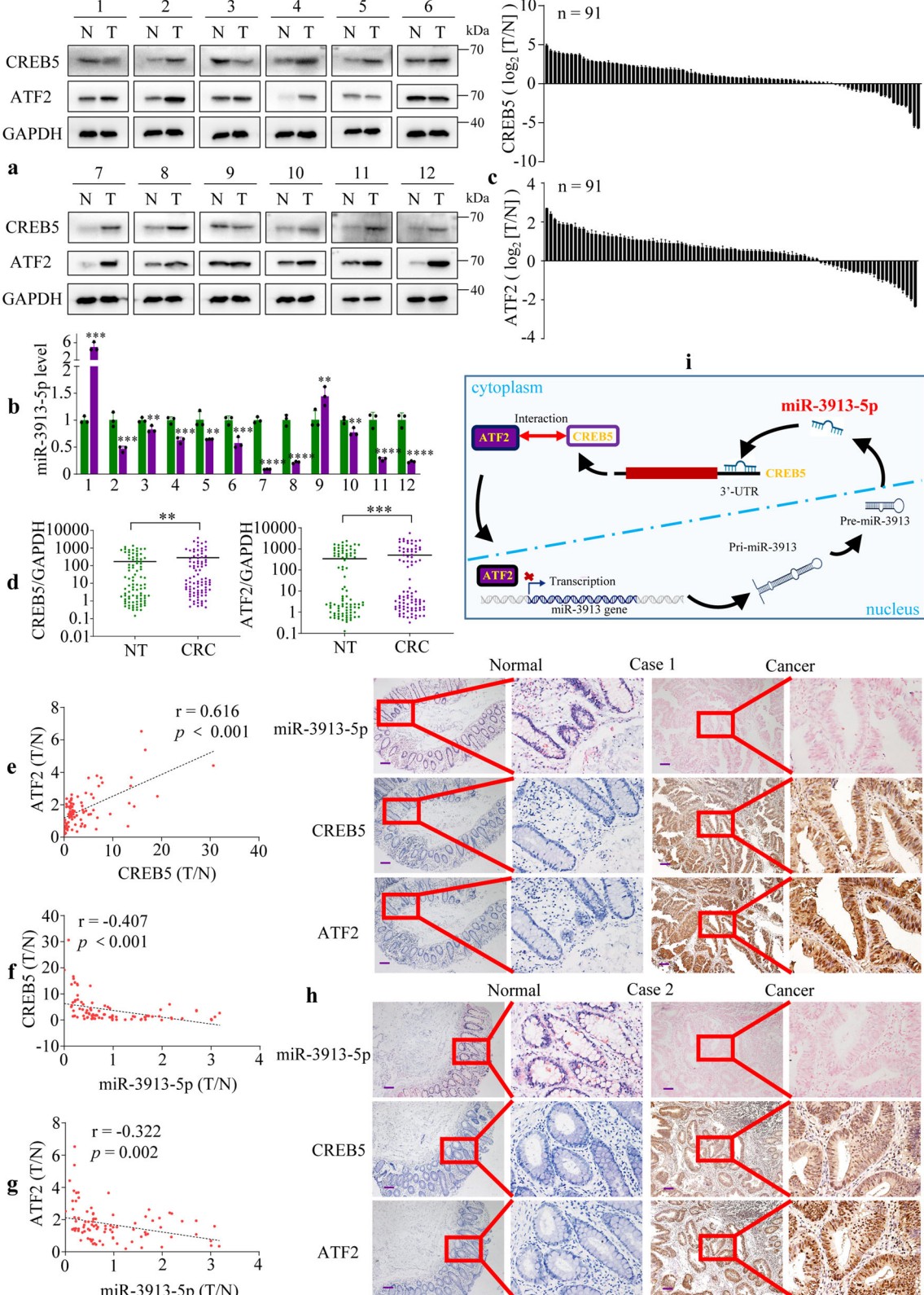

**Fig. 7 miR-3913-5p is negatively correlated with ATF2 and CREB5 expression.** Western blot assays (**a**) and qPCR assays (**b**) showed the CREB5, ATF2 and miR-3913-5p expression in 12 paired CRC tissues and their matched normal tissues. Student's $t$ test; **$p < 0.05$; ***$p < 0.01$; ****$p < 0.001$. **c** QPCR analyses of the CREB5 and ATF2 expression in 91 paired CRC tissues and their adjacent normal tissues. **d** The CREB5 and ATF2 expression levels in 91 pairs of CRC specimens and normal tissues. Student's $t$ test; **$p < 0.05$; ***$p < 0.01$. The expression correlations between CREB5 and ATF2 (**e**), between miR-3913-5p and CREB5 (**f**), between miR-3913-5p and ATF2 (**g**) in CRC tissues. **h** ISH analyses of miR-3913-5p and IHC analyses of CREB5 and ATF2 in the CRC tissues and the corresponding normal tissues. **i** The illustration depicting the mechanism of ATF2/miR-3913-5p/CREB5 axis in CRC. Scale bars, 100 μm in (**h**).

ATF2, belongs to ATF/CREB family, is a multifunctional regulator. Accumulating studies demonstrated ATF2 is implicated in tumor progress. For instance, inhibition of ATF2 substantially decreased Twist-mediated invasion and cell motility of SW480[30]. Downregulation of mir-26b led to upregulation of its target ATF2 and was associated with the poor prognosis in patients with CRC[31,32]. Moreover, ATF2 reversed the decreased metastasis induced by silencing JMJD1C in CRC, suggesting that it might act as an oncogene[33]. In our study, ATF2 was found to be upregulated in CRC by using western blot, qPCR and IHC assays. ATF2 suppression inhibited the proliferation, migration and invasion in CRC cells. And ATF2 overexpression promoted CRC cells growth, migratory and invasive capacities. ATF2 was confirmed to be oncogene in CRC in our study. ATF2 gene also encodes a TF important for both normal development and cancer progression[34,35]. For example, ATF2 bound to the promoter of NEAT1 and the interplay between them contributed to the tumor progression in lung adenocarcinoma[36]. Furthermore, ATF2 interacted directly with the promoter of LINC00882, which exerted the oncogenic functions in hepatocellular carcinoma[37]. In addition, several studies indicated that ATF2 bound to the promoters of various miRNAs, transcriptionally regulating the expression of miRNAs[38,39]. In our study, silencing ATF2 downregulated the expression of miR-3913-5p in CRC cells. Subsequently, ATF2 was verified to be bound to the promoter region of miR-3913-5p by ChIP and luciferase reporter assays. What's more, inhibiting miR-3913-5p weakened the suppression of CRC cells growth, migration and invasion by silencing ATF2 in CRC, suggesting that miR-3913-5p is critical for ATF2 regulating CRC cells proliferation, migration and invasion. Moreover, pearson correlation analysis showed negative correlation between miR-3913-5p and ATF2 in 91 CRC tissues, further confirming ATF2 negatively regulating miR-3913-5p. These studies suggested that ATF2 transrepressed miR-3913-5p expression in CRC cells. Subsequently, we investigated the function of the interaction between CREB5 and ATF2 and found that miR-3913-5p reduced the enhanced proliferation, migration and invasion by ATF2, and CREB5 deteriorated these changes, suggesting that CREB5 was required in ATF2 regulating miR-3913-5p. However, ATF2 was also reported to serve as a tumor suppressor by suppressing the cancer driver TROP2 in CRC[40]. Thus, further experimental researches were needed to achieve the mechanisms of ATF2 in CRC.

Taken together, our findings revealed the downregulation of miR-3913-5p in CRC cell lines and CRC tissues. miR-3913-5p functioned as a tumor suppressor by inhibiting the CRC cells growth and metastasis in vitro and in vivo. Elevated expression of miR-3913-5p led to the downregulation of CREB5 by targeting its 3′UTR. Moreover, we found that ATF2 bound to the promoter of miR-3913-5p to transcriptionally repress miR-3913-5p expression. ATF2 and miR-3913-5p were both upstream regulators of CREB5 and CREB5 was important for ATF2 regulating miR-3913-5p. Thus, our studies demonstrated the critical role of the ATF2/miR-3913-5p/CREB5 axis in CRC, providing valuable insight that might be of use for CRC therapeutic strategies (Fig. 7i).

## Methods

**Cell culture.** The immortalized colonic epithelial cell line FHC, human CRC cell lines RKO and SW1116 were purchased from the American Type Culture Collection (ATCC; Manassas, VA, USA). Human CRC cell lines LoVo, SW480, Caco2 and HT-29 were obtained from the Cell Bank of the Chinese Academy of Sciences (Shanghai, China). All the cell lines were cultured in RPMI-1640 medium (Sigma, St. Louis, MO, USA) containing 10% fetal bovine serum (FBS; Sigma, St. Louis, MO, USA) in a humidified incubator at 37 °C with 5% CO2.

**Tissue specimens.** 91 primary fresh CRC samples, 80 primary fresh GC specimens and the paired noncancerous tissues were collected from Nanfang Hospital, Southern Medical University (Guangzhou, China). Each sample was attached to a confirmed pathological diagnosis and was staged according to the 8th edition of the American Joint Committee on Cancer cancer staging manual. None of the patients has received any preoperative chemo/radiotherapy. The human samples were collected with patients' informed consent and in accordance with the ethical standards of the Medical Ethics Committee of Nanfang Hospital, Southern Medical University. All ethical regulations relevant to human research participants were followed.

**RNA extraction and Quantitative real-time PCR (qPCR).** Total RNA was extracted from cultured cells and frozen fresh tissues by using TRIzol reagent (Invitrogen, Carlsbad, CA, USA). We detected the mRNA expression with the PrimeScript RT Reagent Kit (Takara Bio, Inc., Shiga, Japan) and TB Green Premix Ex Taq (Takara Bio, Inc., Shiga, Japan), and miRNA expressions were detected by All-in-One miRNA qRT-PCR Detection Kits (GeneCopoeia, Inc., Maryland, USA) according to the manufacturer's instructions. U6 and GAPDH were used as miRNA and gene internal controls, respectively. We analyzed the data with the $2^{-\Delta\Delta Ct}$ method. Primers designed for qPCR were provided in Supplementary Table 5.

**In situ hybridization (ISH).** miR-3913-5p expressions in paraffin-embedded CRC tissues and matched normal tissues were detected by ISH. Tissue sections were dewaxed in xylene and rehydrated with a series different gradient ethanol. Then, the tissue sections were treated with 4% paraformaldehyde for 10 min and pepsin dilution in 3% fresh citrate buffer at 37 °C for 30 min. After pre-hybridization at 42 °C for 2 h, hybridization with DIG-labeled miRCURY LNA probes (Exiqon, Vedbaek, Denmark) was performed at 42 °C overnight. Subsequently, the tissue sections were washed with 2 × SSC at 37 °C for 5 min, twice, 0.5 × SSC at 37 °C for 15 min, 0.2 × SSC at 37 °C for 15 min. Then, the sections were treated with blocking solution at 37 °C for 30 min and incubated with anti-DIG antibody at room temperature for 1 h. Hybridization were visualized by nitro blue tetrazolium and 5-bromo-4-chloro-3-indolyl phosphate, and the reaction was stopped by running water. Afterward, the sections were counterstained with nuclear fast red, dehydrated through an ethanol gradient and mounted with aqueous solution. The miR-3913-5p probe sequence was as follows: 5′-AGACATCAAGATCAGTCCCAAA-3′.

**Oligonucleotide and plasmid transfections.** miR-3913-5p mimics, inhibitor, siRNAs for ATF2, ATF7, BATF3, CREB5 and their corresponding controls were designed and synthesized by GenePharma (Shanghai, China). The sequence of CREB5 was constructed into the pcDNA3.1 plasmid vector. The empty vector plasmid was applied as a control (Kidan Bioseiences co., Ltd, Guangzhou, China). We transfected CRC cells with miR-3913-5p mimics, miR-3913-5p inhibitor, siRNA or plasmid using Lipofectamine 3000 reagent (Invitrogen, Carlsbad, CA, USA) according to the manufacturer's protocol. The detailed sequences of the miRNA mimics, miRNA inhibitor and siRNAs were provided in Supplementary Table 6.

**Cell counting Kit-8 (CCK-8) assay.** Cell growth was measured by a Cell Counting Kit-8 (Dojindo, Japan). Cells ($4 \times 10^3$/well) were

seeded into a 96-well plate (Corning, New York, USA) and incubated for 1, 2, 3 or 4 days. Cells were twice rinsed gently with PBS, then we added 10 μL CCK-8 reagent into each well and incubated the cells at 37 °C for 1 h. The absorbance was determined at a wavelength of 450 nm.

**Colony formation assay**. Cells were seeded in 12-well plates at a density of 100 cells/well. After incubated at 37 °C for 2 weeks, the cells were washed with PBS, fixed with 4% paraformaldehyde for 20 min and stained with a 0.5% crystal violet solution for 5 min. The colonies were examined and counted under a microscope.

**EdU incorporation assay**. Cell-Light EdU Apollo567 In Vitro Kit (RIBOBIO, Guangzhou, China) was used to examine cell proliferation according to the standard protocol. $3 \times 10^4$ cells were seeded in a 96-well plate and incubated overnight. Then, the cells were incubated with 50 μM EdU for 2 h, fixed with 4% paraformaldehyde and permeabilized with 0.5% Triton X-100. Subsequently, $1 \times$ Apollo reaction cocktail was added to react with the EdU. Nuclei were visualized with Hoechest33342. EdU-positive cells were counted under an inverted fluorescence microscope Olympus IX73 (Olympus Corporation, Tokyo, Japan).

**Transwell assay**. For migration assays, cells were resuspended in serum-free PRMI-1640 medium. Then, $4 \times 10^4$ cells were incubated in the transwell chambers with 8-μm pores (Corning, New York, USA). For invasion assays, $8 \times 10^4$ cells were placed into the chambers pre-coated with Matrigel (BD Biosciences, New Jersey, USA). Then, PRMI-1640 with 10% FBS was added to a 24-well plate. The chambers were inserted into the 24-well plate and incubated for 48 h at 37 °C. Cotton swabs were used to remove upper chamber cells and cells stuck to the low surface were fixed with the 4% paraformaldehyde and stained with 0.1% crystal violet. Finally, migrating cells were counted under an inverted microscope.

**Wound healing assay**. Cells were seeded into 6-well plate and incubated until they reached 80% confluence. A scratch wound was artificially created by using a 200 μL pipette tip. We used PBS to remove the floating cells and photographed the scratches under an inverted microscope. The diminishing distance across the scratch wound was measured and expressed as a relative migration rate, normalized to the original scratch width.

**Western blot**. Protein was extracted from cells or tissues by using RIPA buffer and quantified by the BCA method. Subsequently, protein was electrophoresed on sodium dodecyl sulfate-polyacrylamide gel electrophoresis, and then transferred to polyvinylidene fluoride membranes followed by incubation with 5% nonfat powdered milk in Tris-buffered saline at room temperature for 1 h. Then, the membranes were incubated with the specific primary antibodies overnight at 4 °C. The next day, after washed with TBST, membranes were incubated with secondary antibodies at room temperature for 1 h and washed again. The immunoreactive bands were visualized by an enhanced chemiluminescence system (Millipore, Billerica, MA, USA). The primary antibodies were as follows: anti-CREB5 (Santa Cruz, sc-130435), anti-ATF2 (Immunoway, YT0382) antibodies. GAPDH (Proteintech, 60004-1-Ig) were used as an internal control.

**Chromatin immunoprecipitation (ChIP) assay**. Cells were cross-linked with 1% formaldehyde. DNA fragments ranging from 400 to 500 bp were yielded via sonication and subjected to the immunoprecipitation process with anti-ATF2 antibody (Cell Signaling, #35031). The detailed processes were performed

following the protocol provided by the SimpleChIP Enzymatic Chromatin IP kit with magnetic beads (#9003, Cell Signaling, Danvers, MA, USA). Finally, PCR was performed to examine the enrichment of DNA fragments in the putative ATF2 binding sites of the promoter of miR-3913-5p with the specific primers (Supplementary Table 5).

**Luciferase reporter assay**. For the miRNA binding site assays, CREB5 3'UTR sequences containing the wide-type predicted microRNA binding sites (WT1, WT2) or mutant ones (MT1, MT2) were cloned into pmirGlo vectors (Kidan Bioseiences co., Ltd, Guangzhou, China). The empty vector was used as the control. Wide-type vectors, mutant vectors or the control vector were co-transfected into CRC cells with miR-3913-5p mimics or m-NC. For the MIR3913-1 or MIR3913-2 promoter activity assays, MIR3913-1 or MIR3913-2 promoter regions containing predicted binding sites of ATF2 were amplified and inserted into the pGL3-Basic vectors (Kidan Bioseiences co., Ltd, Guangzhou, China). These pGL3-Basic-derived vectors and ATF-2-expressing vector were co-transfected into CRC cells. Luciferase reporter assays were performed at 48 h after transfection by the Dual-Luciferase Reporter Assay System (Promega, Madison, WI, USA) according to the manufacturer's protocols.

**Co-immunoprecipitation (Co-IP) assay**. About 90% adherent cells were lysed with cold cell lysis buffer or 30 min at 4 °C and centrifuged at 12,000 rpm for 20 min at 4 °C. Protein supernatant was incubated with specific antibody (anti-CREB5: Santa Cruz, sc-130435; anti-ATF2: Immunoway, YT0382) or IgG control overnight at 4 °C. Subsequently, protein A/G agarose beads were added to the immunoprecipitation mixture with gentle rocking for 2 h at 4 °C. After washing with washing buffer, the beads with the immunoprecipitates were incubated with $1 \times$ sample loading buffer and boiled for 10 min for subsequent western blot analysis.

**Immunohistochemistry (IHC)**. Paraffin-embedded tissue sections were baked at 65 °C for 2 h, dewaxed with xylene and hydrated with gradient ethanol. We performed the antigen retrieval by microwave in sodium citrate buffer (pH 6.0). Next, tissues sections were immersed in 3% $H_2O_2$ to block endogenous peroxidase activity and blocked for 1 h in sealing liquid conjugated with 5% bovine serum albumin. Tissue sections were incubated with primary antibodies (anti-CREB5: abcam, ab168928; anti-ATF2: Immunoway, YT0382) at 4 °C overnight. The next day, the slices were rewarmed and washed. Afterward, tissue sections were treated with secondary antibody at room temperature for 1 h. Visualization was performed with 3,3'-diaminobenzidine tetrahydrochloride chromogen. And the sections were counterstained with hematoxylin, dehydrated and sealed with neutral balsam. The staining results were scored according to the carcinoma cell staining intensity as follows: 0, negative staining; 1, weak staining; 2, moderate staining; and 3, strongly staining. The average score for each sample was evaluated independently by two pathologists. Negative and weak staining were considered as low expressed, and moderate and strongly staining were considered to be high expressed.

**Lentivirus production and infection**. The lentiviral vector (Ubi-MCS-SV40-Cherry) expressing miR-3913-5p containing the red fluorescent protein gene was constructed by GeneChem (Shanghai, China). The empty vector was used as the control. CRC cells were infected with the lentiviral vector and the infection efficiency was measured by qPCR.

**Animal models**. All 4- to 6-week-old BALB/c nu/nu nude female mice used in our study were maintained in a specific-pathogen free (SPF) environment at Laboratory Animal Unit, Southern Medical University (Guangzhou, China). For in vivo growth assay, $1 \times 10^7$ miR-3913-5p overexpressing CRC LoVo cells or the control CRC cells resuspended in 0.1 ml PBS were subcutaneously injected into the right flank of each nude mouse. We examined the sizes of the resulting tumors every 3 days with a vernier caliper, using the formula: $V = L \times W^2/2$, where V is the volume, L is the length and W is the width. At 42 days after inoculation, the mice were sacrificed and their tumors were resected and imaged using the In Vivo Imaging System (FX PRO, Bruker, Billerica, MA, USA). For in vivo metastasis assay, $5 \times 10^6$ miR-3913-5p overexpressing LoVo cells or their controls were injected into the tail vein of each nude mouse. After 50 days, all mice were sacrificed and their lung tissues were collected for examination. The metastatic loci was calculated by combining visual observation and fluorescence imaging. The animal experiments were conducted in accordance with institutional guidelines and approved by the Southern Medical University Experimental Animal Ethics Committee. We have complied with all relevant ethical regulations for animal testing.

**Statistics and reproducibility**. Data were analyzed using IBM SPSS Statistics 23 (SPSS, Inc., Chicago, IL, USA) and GraphPad Prism 8 (GraphPad Software, San Diego, CA, USA) and expressed as mean ± SD with at least three independent experiments. The Student's $t$ test or one-way ANOVA was used to evaluate the significance of data from qPCR assays, proliferation assays, migration and invasion assays, in vivo assays and luciferase reporter assays. Survival analyses were performed using Kaplan–Meier and log-rank tests. Correlations between genes and miRNA were determined by Pearson correlation. $P < 0.05$ was considered to be statistically significant.

**Reporting summary**. Further information on research design is available in the Nature Portfolio Reporting Summary linked to this article.

## Data availability
All data generated or analyzed during this study are included in the article and supplemental files, or available from the corresponding author on reasonable request. All source data underlying the graphs and charts showed in the figures are presented in Supplementary Data File 1. Uncropped and unedited blots/gels are presented in Supplementary Figs. 8, 9.

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

## Acknowledgements

We acknowledge the generous support of the Guangdong Provincial Key Laboratory of Gastroenterology, Department of Gastroenterology, Nanfang Hospital, Southern Medical University and the Department of Gastroenterology, Longgang District People's Hospital, Shenzhen. This work was supported by the National Natural Science Foundation of China (Grant/Award number: 82273354, 82302891, 82372955, 81974448, 82103152, 82103598, 82073066), Guangdong Basic and Applied Basic Research Foundation (Grant/Award number: 2020A1515110059), Natural Science Foundation of Guangdong Province (Grant/Award number: 2022A1515012464), General guide project of Health Science and technology in Guangzhou (Grant/Award number: 20221A011120), Shenzhen Science and Technology Innovation Commission Fund (Grant/Award number: JCYJ20210324135005013), Longgang District Science and Technology Innovation Bureau (Grant/Award number: LGKCYLWS2021000012, LGKCYLWS2022-005), Hospital and School Joint Fund (Grant/Award number: YXLH2209).

## Author contributions

J.D.W., M.M.P. and J.Y.L. designed and conceived this study. W.Y.D., L.J.H., W.S.X., L.Y.Z., M.M.P. and Z.Y. performed the experiments in vitro. Experiments in vivo were performed by W.Y.D., L.J.H., X.H.L., S.D.L., Y.Z.X., P.Y. Tissue samples were collected by W.M.T., Z.Z.L. Y.P., J.M.Z. contributed to data analysis and interpretation. J.J.L., X.S.W., W.H.S. performed the statistical analysis. L.X., W.M.T. helped with technical support. S.D.L. and X.S.W. coordinated the project. L.J.H. and Z.Z.L. supervised the project. W.Y.D. wrote the manuscript and L.J.H. revised the manuscript. All authors read and approved the final manuscript.

## Competing interests

The authors declare no competing interests.
