## [Peer Review File · Communications Biology]

Reviewers' comments:

Reviewer #1 (Remarks to the Author):

The authors describe a novel ATF2/miR-3913-5p/CREB5 regulatory axis in colorectal cancer cells for cell growth, migration, and invasion. CREB5 has been identified as a direct target of miR-3913-5p. Moreover ATF2 has been shown to target miR-3913-5p. This has been mechanistically confirmed by ATF2 si transfection, luciferase reporter assays and CHIP experiments. The functional role of this triple axis was shown very convincingly in vitro and in vivo. ATF2 has been identified as an oncogene being associated with worse prognostic parameters.

The manuscript is well structured and has impressive images and Figures. The methodological panel is state-of-the art and well described. Nevertheless there are a few flaws and criticisms. The authors need to address these points to improve the quality of the manuscript.

Although the role of ATF2 in colorectal cancer is discussed controversially the authors should discuss at least a few reasons why their study confirmed an oncogenic role of ATF2. They could discuss the composition of the tumor group, the used panel of cell lines, the test models, the problem of AP1 binding partners Etc.

1. Co-immunoprecipitation experiment: they show that ATF2 and CREB5 built a complex in SW480 and LoVo cells. How is the complex in the miR-3913-5p high expressing normal FHC cells? Are ATF7 and BATF3 in the complex with CREB5?

2. the in vivo experiment: The authors performed a xenograft experiment but they do not really take advantage of this experiment to verify the triple axis in vivo. How is immunostaining for CREB5 to verify the inhibitory role of miR-3913-5p in vivo? Although ATF2 is upstream of miR, how is ATF2 staining pattern?

Figure 2h: the authors show that xenografts are smaller when overexpressing miR-3913-5p. How are they sure that this is only an effect of less proliferation and not apoptosis induction. The authors should provide HE slices and examine at least the growth pattern of xenografted tumors. Ki67 staining could give a clue about proliferation stop. They could also provide apoptosis data in vitro. In Figure 3g, h they show the macroscopic results of the tail vein injection experiment: The authors need to give more details of the procedure: how they counted the metastatic loci? Did they measure the weight of the lungs? Is n=3 animals per group enough to perform reliable statistics?

In the methods part line 410 the authors give siRNA for c-JUN, but they do not present data using c-JUN si. It would make sense to test if c-JUN is the partner of ATF2 in binding to the promoter of miR. Since CREB5 is interacting with ATF2 it might also play a role in AP1- forming complex. This could be shown by CHIP for CREB5. Then the triple axis would be a positive regulatory loop to decrease the miR-3913-5p expression, a fact that would remarkably increase the attractiveness of this manuscript.

Reviewer #2 (Remarks to the Author):

Dai et al present a study “The ATF2/miR-3913-5p/CREB5 axis is involved in the cell proliferation and metastasis of colorectal cancer”. They have examined the consequences of miR-3913-5p down regulation in CRC cells, in vivo models, and patient samples. Extensive laboratory and informatics approaches come together to demonstrate that ATF2 regulates mi-3913-5p expression, which in turn regulates the CREB5 UTR sequence. This relationship is relevant for several critical cellular functions and appears to hold true in patients. In general, the experiments are elegant and well designed and the manuscript is well written. The basis of the observations are generally well supported.

Comments:

1. The authors indicate that CREB5 and ATF2 directly interact with one another based on Co-IP experiments (Figure 5c). The function of this binding was not thoroughly discussed in the Title, Figure 7i and insufficiently discussed in the Abstract and Discussion. In the Abstract, it is stated that “CREB5 could cooperate with ATF2.” In Figure 7i, it is unclear if the interaction is functional. Is CREB5 required for ATF2 in regulating miR-3913-5p? Or does CREB5 have separate functions with ATF2 that do not regulate miR-3913-5p?
2. In addition, the title gives the readers the notion that ATF2 and miR-3913-5p are both upstream regulators of CREB5 (also in Figure 7i). Once the authors define the relationship in each section, they should update each section accordingly.
3. In Figure 7h, it is difficult to examine if CREB5 is co-expressed with ATF2 in the same cells. Some simple quantification and statistical test of the expression would be of value to the readers.
4. The labeling of the figure panels can be confusing and unconventional at times (Figure b1, b2, etc.).
5. The discussions of CREB5 and FOXA1 interactions should include the citation PMID: 35550030

Rebuttal Letter

The ATF2/miR-3913-5p/CREB5 axis is involved in the cell proliferation and metastasis of colorectal cancer

Weiyu Dai et al.

Reviewer #1 (Remarks to the Author):

The authors describe a novel ATF2/miR-3913-5p/CREB5 regulatory axis in colorectal cancer cells for cell growth, migration, and invasion. CREB5 has been identified as a direct target of miR-3913-5p. Moreover ATF2 has been shown to target miR-3913-5p. This has been mechanistically confirmed by ATF2 si transfection, luciferase reporter assays and ChIP experiments. The functional role of this triple axis was shown very convincingly in vitro and in vivo. ATF2 has been identified as an oncogene being associated with worse prognostic parameters.

The manuscript is well structured and has impressive images and Figures. The methodological panel is state-of-the art and well described. Nevertheless there are a few flaws and criticisms. The authors need to address these points to improve the quality of the manuscript.

Although the role of ATF2 in colorectal cancer is discussed, controversies the authors should discuss at least a few reasons why their study confirmed an oncogenic role of ATF2. They could discuss the composition of the tumor group, the used panel of cell lines, the test models, the problem of AP1 binding partners Etc.

Reply: Thank the reviewer for the constructive comments and advice on our study. In our study, ATF2 was confirmed to act as an oncogenic role in CRC. The reasons are listed as follows. First, ATF2 was found to be upregulated in CRC by using western blot, qPCR and IHC assays (Fig. 7a, c, d, h, Supplementary Fig. 5a). Interference of ATF2 inhibited the proliferation, migration and invasion in LoVo and SW480 cells (Fig. 6a-d, Supplementary Fig. 7a-d). And ATF2 overexpression promoted cell growth, migratory and invasive capacities in CRC cells (Supplementary Fig. 6a-d). What's more, ATF2 interference could increase miR-3913-5p expression, which then suppressed CREB5 and inhibited the proliferation and metastasis of CRC (Fig. 2, 3, 4c, 5b). So ATF2 was found to be oncogene in CRC in our study. We had added some detailed reasons why we confirmed an oncogenic role of ATF2 in the part of discussion. The above modifications were colored in red. What's more, we addressed the other issues that the reviewer concerned in the responses below.

1. Co-immunoprecipitation experiment: they show that ATF2 and CREB5 built a complex in SW480 and LoVo cells. How is the complex in the miR-3913-5p high expressing normal FHC cells? Are ATF7 and BATF3 in the complex with CREB5?

Reply: Many thanks for your comment. We performed the corresponding co-IP assays showing that ATF2 and CREB5 built a complex in FHC cells (Reviewers' comments. 1a). But in CRC cells (LoVo or SW480 cells), miR-3913-5p, CREB5 and ATF2 expressions might be different from normal cells, influencing the CRC progression. What's more, co-IP assays suggested that CREB5 cooperated with ATF7 or BATF3 in LoVo and SW480 cells (Reviewers' comments. 1b). However, the interference of ATF7 or BATF3 could not change the miR-3913-5p expression while the knockdown of ATF2 markedly increase the miR-3913-5p expression (Fig. 5b), so we explored how ATF2

effected miR-3913-5p in this manuscript.

2. the in vivo experiment: The authors performed a xenograft experiment but they do not really take advantage of this experiment to verify the triple axis in vivo. How is immunostaining for CREB5 to verify the inhibitory role of miR-3913-5p in vivo? Although ATF2 is upstream of miR, how is ATF2 staining pattern?

Reply: Thank you for your suggestion. We made the ISH staining of miR-3913-5p and IHC staining of CREB5 and ATF2 of subcutaneous tumors, identifying the higher expression of CREB5 and ATF2 in the subcutaneous tumors with low miR-3913-5p expression in Lv-NC group compared with Lv-miR-3913-5p group (Supplementary Fig. 5c). We have made necessary revision in the section of “Results” and colored it in red.

Figure 2h: the authors show that xenografts are smaller when overexpressing miR-3913-5p. How they are sure that this is only an effect of less proliferation and not apoptosis induction. They could also provide apoptosis data in The authors should provide HE slices and examine at least the growth pattern of xenografted tumors. Ki67 staining could give a clue about proliferation stop vitro.

Reply: We appreciate the reviewer’s comment. In Supplementary Fig. 2a, Apoptosis inductions were observed by Hoechst 33258 staining in LoVo and SW480 cells with miR-3913-5p increased in vitro. We performed the HE and IHC staining of Ki-67 or Cleaved caspase-3 in subcutaneous tumors of mice. In Supplementary Fig. 2b, lower expression of Ki-67 and higher cleaved caspase-3 were found in the Lv-miR-3913-5p groups, showing less proliferation and apoptosis induction existed simultaneously in Lv-miR-3913-5p groups in vivo. What’s more, HE and IHC staining of MMP2 in lungs of mice were conducted (Supplementary Fig. 3), showing less metastasis in the Lv-miR-3913-5p groups. We have made the supplements and colored in red. Thanks.

In Figure 3g, h they show the macroscopic results of the tail vein injection experiment:

The authors need to give more details of the procedure: how they counted the metastatic loci? Did they measure the weight of the lungs? Is n=3 animals per group enough to perform reliable statistics?

Reply: Thank you for your comment. More details about counting the metastatic loci were added and colored in red in the Methods part. We didn't measure the weight of the lungs in our study. We thought that each nude mouse have different weight, so the weight of their lungs could not represent the metastasis in lungs. In Fig. 2h, i, we made the replication of the subcutaneous tumors model with six nude mice in each groups, showing the smaller tumors in the Lv-miR-3913-5p groups. What's more, we had performed the in vivo metastasis assay with five nude mice before. The modifications were added in Fig.3g, h, suggesting that less metastatic lung nodules in the Lv-miR-3913-5p groups.

In the methods part line 410 the authors give siRNA for c-JUN, but they do not present data using c-JUN si. It would make sense to test if c-JUN is the partner of ATF2 in binding to the promoter of miR. Since CREB5 is interacting with ATF2 it might also play a role in AP1- forming complex. This could be shown by ChIP for CREB5. Then the triple axis would be a positive regulatory loop to decrease the miR-3913-5p expression, a fact that would remarkably increase the attractiveness of this manuscript.

Reply: Thank you for your constructive and insightful advice. In the methods part, "siRNAs for c-JUN" was a mistake in writing. We apologize for our carelessness. The hypothesis of the reviewer is interesting and deeply enlighten us. We will try to test if c-JUN is the partner of ATF2 in binding to the promoter of miR-3913 and explore the role of CREB5 in AP-1 forming complex in our next research subject. We deleted the word "c-JUN". Thanks for your comments again.

Reviewer #2 (Remarks to the Author):

Dai et al present a study "The ATF2/miR-3913-5p/CREB5 axis is involved in the cell proliferation and metastasis of colorectal cancer". They have examined the consequences of miR-3913-5p down regulation in CRC cells, in vivo models, and patient samples. Extensive laboratory and informatics approaches come together to demonstrate that ATF2 regulates mi-3913-5p expression, which in turn regulates the CREB5 UTR sequence. This relationship is relevant for several critical cellular functions and appears to hold true in patients. In general, the experiments are elegant and well designed and the manuscript is well written. The basis of the observations are generally well supported.

Comments:

1. The authors indicate that CREB5 and ATF2 directly interact with one another based on Co-IP experiments (Figure 5c). The function of this binding was not thoroughly

discussed in the Title, Figure 7i and insufficiently discussed in the Abstract and Discussion. In the Abstract, it is stated that “CREB5 could cooperate with ATF2.” In Figure 7i, it is unclear if the interaction is functional. Is CREB5 required for ATF2 in regulating miR-3913-5p? Or does CREB5 have separate functions with ATF2 that do not regulate miR-3913-5p?

Reply: Thank the reviewer for the positive comments on the strength of our study and the insightful advice. To explore the role of CREB5 in ATF2 regulating miR-3913-5p, investigating the function of the CREB5 and ATF2 interaction, we performed the colony formation, EdU and the transwell assays. In Supplementary Fig. 6a-d, the colony formation, EdU and transwell assays showed that miR-3913-5p overexpression weakened the enhanced proliferation, migration and invasion by ATF2, and CREB5 overexpression could reverse these effects in LoVo and SW480 cells. In Supplementary Fig. 7a-d, it was found that miR-3913-5p suppression eroded the weakened cell growth, migratory and invasive capacities of CRC cells by ATF2 inhibition, while CREB5 inhibition deteriorated the effects above. So CREB5 was required for ATF2 in regulating miR-3913-5p. We made some revisions in the Abstract, Results and Discussion to describe that CREB5 was required for ATF2 in regulating miR-3913-5p.

2. In addition, the title gives the readers the notion that ATF2 and miR-3913-5p are both upstream regulators of CREB5 (also in Figure 7i). Once the authors define the relationship in each section, they should update each section accordingly.

Reply: Thank you for your suggestion. We have checked our manuscript and updated in the Results, and Discussion section. The revisions were marked in red.

3. In Figure 7h, it is difficult to examine if CREB5 is co-expressed with ATF2 in the same cells. Some simple quantification and statistical test of the expression would be of value to the readers.

Reply: We appreciate the reviewer’s comment. We provided the IHC of CREB5 and ATF2 and ISH of miR-3913-5p of other 13 CRC tissues and their matched normal tissues in Supplementary Fig. 5a. And we made the simple quantification and statistical test of the IHC stainings of CREB5 and ATF2 in the tissues, showing the positive correlation between CREB5 and ATF2 (Supplementary Fig. 5b).

4. The labeling of the figure panels can be confusing and unconventional at times (Figure b1, b2, etc.).

Reply: Thank you for your carefulness and we apologize for making the confusion. We revised the labeling and colored it in red.

5. The discussions of CREB5 and FOXA1 interactions should include the citation PMID: 35550030

Reply: Thanks for your comment. The roles of CREB5 and FOXA1 interactions in resistance to androgen receptor-targeting therapies enlightened us that CREB5 interacting with other transcriptional factors might be important in cancer development. We included the citation PMID: 35550030 in the part of Discussion and colored the text in red.

REVIEWERS' COMMENTS:

Reviewer #1 (Remarks to the Author):

I appreciate very much how thoroughly the authors answered my questions. They performed meaningful additional experiments and finally improved the quality of the manuscript significantly.

Figure 6: no scale bars in C) and D)?

Reviewer #2 (Remarks to the Author):

The authors have fully addressed the original critiques.

Rebuttal Letter

The ATF2/miR-3913-5p/CREB5 axis is involved in the cell proliferation and metastasis of colorectal cancer

Weiyu Dai et al.

Reviewer #1 (Remarks to the Author):

I appreciate very much how thoroughly the authors answered my questions. They performed meaningful additional experiments and finally improved the quality of the manuscript significantly.

Figure 6: no scale bars in C) and D)?

Reply: Thank you for your comments. We have added the scale bars in Fig. 6c and d. We appreciate your insightful advice.

Reviewer #2 (Remarks to the Author):

The authors have fully addressed the original critiques.

Reply: Thank you for your recognition very much.